# Dichloroacetate and Quercetin Prevent Cell Proliferation, Induce Cell Death and Slow Tumor Growth in a Mouse Model of HPV-Positive Head and Neck Cancer

**DOI:** 10.3390/cancers16081525

**Published:** 2024-04-17

**Authors:** Yongxian Zhuang, Joseph D. Coppock, Allison B. Haugrud, John H. Lee, Shanta M. Messerli, W. Keith Miskimins

**Affiliations:** Cancer Biology and Immunotherapies, Sanford Research, Sioux Falls, SD 57104, USA; yongxian.zhuang@gmail.com (Y.Z.); jcoppock@pmlpathology.com (J.D.C.); allison.haugrud@gmail.com (A.B.H.); john.lee@avera.org (J.H.L.)

**Keywords:** Dichloroacetate (DCA), quercetin, human papilloma virus, head and neck oral cancer, lactate

## Abstract

**Simple Summary:**

The metabolism of cancer cells and the tumor microenvironment are of increasing interest as part of ongoing efforts to develop potential adjuvant therapies to be used along with conventional chemotherapy and radiation. In this report, the antitumor properties of two compounds that affect glucose metabolism, dichloroacetate and quercetin, are examined. Both DCA and quercetin, a naturally occurring plant flavonoid found in fruits and vegetables, demonstrate inhibitory effects on the growth of head and neck cancer both in cell culture and in a preclinical mouse model of head and neck cancer. These two compounds have synergistic antitumor effects when combined, both in vitro and in vivo. The drug combination inhibited tumor growth and induced cell death with the maintenance of an unfavorable tumor microenvironment. The altered tumor microenvironment appears to enhance immune-mediated clearance of tumors. Thus, this study supports additional preclinical research to further explore the antitumor effects of DCA and quercetin.

**Abstract:**

Elevated glucose uptake and production of lactate are common features of cancer cells. Among many tumor-promoting effects, lactate inhibits immune responses and is positively correlated with radioresistance. Dichloroacetate (DCA) is an inhibitor of pyruvate dehydrogenase kinase that decreases lactate production. Quercetin is a flavonoid compound found in fruits and vegetables that inhibits glucose uptake and lactate export. We investigated the potential role and mechanisms of DCA, quercetin, and their combination, in the treatment of HPV-positive head and neck squamous cell carcinoma, an antigenic cancer subtype in need of efficacious adjuvant therapies. C57Bl/6-derived mouse oropharyngeal epithelial cells, a previously developed mouse model that was retrovirally transduced with HPV type-16 E6/E7 and activated Ras, were used to assess these compounds. Both DCA and quercetin inhibited colony formation and reduced cell viability, which were associated with mTOR inhibition and increased apoptosis through enhanced ROS production. DCA and quercetin reduced tumor growth and enhanced survival in immune-competent mice, correlating with decreased proliferation as well as decreased acidification of the tumor microenvironment and reduction of Foxp (+) Treg lymphocytes. Collectively, these data support the possible clinical application of DCA and quercetin as adjuvant therapies for head and neck cancer patients.

## 1. Introduction

Head and neck cancer accounts for three to five percent of all cancers in the United States, of which smoking and alcohol use are the leading causes [1,2,3]. However, human papilloma virus (HPV) infection, especially HPV type-16 (HPV-16), has been a rapidly increasing cause of head and neck cancer, especially of the oropharynx [4,5]. Researchers believe that 60–80% of oropharyngeal cancers are caused by HPV [5]. Even though most HPV-positive (HPV+) head and neck cancers are diagnosed at advanced stages, they tend to respond to the current treatments of chemotherapy and radiation better than head and neck cancers caused by smoking and alcohol [6,7]. Previous studies have shown that this is not because HPV+ cancer cells are more sensitive to chemoradiation treatment, but rather that it involves the antigenic properties of the tumor that promote an immune response, allowing for tumor clearance [8]. Due to the importance of the immune response during chemoradiation therapy of HPV+ head and neck cancer, compounds that enhance the immune response, or eliminate factors that disrupt the immune response, could be potential adjuvant therapies during conventional chemoradiation treatment.

It is a common feature of cancer cells to have increased glucose uptake and accumulation of the metabolic byproduct, lactate, even under normoxic conditions [9,10]. High glycolytic flux in cancer cells yields ATP and biosynthetic intermediates, which provide anabolic biomass necessary for tumor growth. In this process, cancer cells convert pyruvate primarily into lactate rather than delivering it to mitochondria for oxidation [11]. Lactate plays a key role in tumor development and growth, in escaping immune surveillance, and in metastasis [12,13]. Lactate has also been shown to induce the secretion of VEGF, promoting new blood vessel formation to ensure sufficient oxygen and nutrient supply for tumor proliferation [14]. Furthermore, lactate has been demonstrated to induce the migration of cells and cell clusters and to induce the secretion of hyaluronan by tumor-associated fibroblasts, aiding tumor metastasis [12,15,16]. Several publications have documented the ability of lactate to suppress the immune system, preventing clearance of tumors [16,17,18] Clinically, tumor lactate levels have great value in predicting metastases and overall survival in cancer patients and have been positively correlated with radioresistance [19]. In summary, the accumulation of lactate in solid tumors plays a pivotal role not only in the development of malignancies but also in resistance to chemoradiation and in tumor recurrence through multiple aspects.

Dichloroacetate (DCA), an inhibitor of pyruvate dehydrogenase kinase (PDK), decreases lactate production through the disinhibition of pyruvate dehydrogenase, promoting the conversion of pyruvate to acetyl CoA instead of lactate. DCA has been shown to increase reactive oxygen species (ROS) production, lead to cancer cell death [20,21], and alter tumor pH [22]. Several clinical trials have been conducted using DCA as an adjuvant drug for cancer treatment, including head and neck cancer. Phase II clinical trials demonstrated that combining DCA with cisplatin-based chemoradiation therapy (CRT) in head and neck cancer patients was safe with no negative effects on survival but had significantly higher end-of-treatment response rates compared to cisplatin-based CRT alone [23]. Our previous data suggested that DCA alone did not have a significant role in modulating cell proliferation and cell death in HPV-16 E6/E7 and Ras-transformed mouse oropharyngeal epithelial (MEER) cells in a previously established mouse model of HPV+ head and neck squamous cell cancer (HNSCC) [24]. However, DCA increased the clearance of tumors in combination with standard-of-care chemoradiation in immune-competent mice, though not in immune-compromised mice [17]. This suggests that modulation of the immune response to the tumor is at play, possibly through reduction of tumor lactate production.

Quercetin is a flavonoid compound found in many kinds of fruits and vegetables. It has been shown to inhibit several kinases that are involved in mitotic processes [25]. Numerous reports have also demonstrated that quercetin is an effective natural inhibitor of cancer invasion and metastasis [26,27,28]. Of interest, quercetin has been shown to be involved in the regulation of glucose uptake and lactate transport as well as in the regulation of cell growth and cell death [29,30,31,32]. Quercetin decreases glucose uptake, and this could reduce the production of tumor cell ATP as well as biosynthetic intermediates for anabolic reactions [33]. Quercetin has also been found to increase ROS production, and thus could have synergistic effects with DCA and radiation therapy by further increasing ROS levels and DNA damage, leading to cell death [34,35,36].

Quercetin regulates lactate levels through its effects on lactate transporters. It is considered to be an inhibitor of lactate transporters, leading to the reduction of lactate transport from the inside of the cell to the extracellular tumor microenvironment [32]. In this study, we evaluate the hypothesis that quercetin and DCA will synergize to inhibit cell proliferation, increase ROS production, induce cell death, and enhance tumor clearance in the face of attenuated lactate accumulation in the tumor microenvironment. This work will provide supporting evidence for future clinical testing of the combination of DCA and quercetin for HPV+ HNSCC patients.

## 2. Materials and Methods

### 2.1. Chemicals and Cell Culture

DCA and quercetin were purchased from Sigma (St. Louis, MO, USA). Mouse MEER cells were provided by Dr. John Lee and were previously internally derived from C57Bl/6 mouse oropharyngeal epithelium (MOE) through retroviral transduction as previously described [37]. Human cell lines were acquired from ATCC. All cell lines have been validated through STR testing and shown to be free of pathogens via RADIL testing.

### 2.2. Colony Formation Assay

MEER cells were seeded on 100 mm plates (500 cells per plate) in triplicate. After one day, cells were treated as indicated and allowed to grow until colonies of untreated controls reached 50 or more cells (7 days). Cells were then fixed with 70% ethanol and stained with Coomassie blue. Imaging was carried out using an Alpha Imager (Alpha Innotech, San Leandro, CA, USA) low light imaging system, and colonies of more than 15 cells were counted using the provided software (ChemImager 4000 (for Windows Version 3.3b.).

### 2.3. Sytox Green Cytotoxicity Assay

MEER cells were plated on 96-well plates. The next day cells were treated as indicated for an additional day. Sytox green nucleic acid stain (Invitrogen (Carlsbad, CA, USA)) was then added to each well at a concentration of 10 µM for 10 min, and plates were then read at 485/580 with a 515 cutoff filter using a SpectraMax M5 fluorescent plate reader (Molecular Devices, San Jose, CA, USA. The first reading is indicative of the number of dead cells. All cells were then permeabilized using 0.4% Triton X-100 for 10 min and plates were read again with the same settings. The second reading is indicative of total cell number, which together with the first reading was then used to calculate a percentage of dead cells.

### 2.4. Cell Proliferation Assay

MEER cells were plated on 35mm dishes. The next day cells were treated as indicated at the indicated concentrations of the drug for one additional day. MEER cells were then trypsinized and stained with trypan blue (0.2%). Live cells, which excluded trypan blue, were counted via hemacytometer. Human head and neck HPV+ cancer cell lines were treated with DCA (10 mM), quercetin (25 µM), or the combination of DCA and quercetin for one day on 96-well plates, and the percentage of dead cells was measured using Sytox Green Staining (ThermoFisher Scientific, Waltham, MA, USA)

### 2.5. Western Blotting

MEER cells and human head and neck HPV+ cancer cell lines were treated with DCA (10 mM), quercetin (25 µM), or the combination of DCA and quercetin for one day on 96-well plates, and Western blotting was performed to detect PARP and cleaved PARP with same treatments. Cells were harvested in an SDS-containing buffer (2.5 mmol/L Tris–HCl (pH 6.8), 2.5% SDS, 100 mmol/L dithiothreitol, 10% glycerol, 0.025% pyronine Y) at 24 hr., sonicated, and heated before running Western blots. Equal amounts of protein were separated via SDS-PAGE and transferred to the PVDF membrane via a semi-dry transfer apparatus. Antibodies for detection of proteins of interest were diluted in blocking buffer, 5% BSA or non-fat dry milk in TBST, as recommended for individual antibodies: Phospho-p70 S6 Kinase (Thr389, #9206, Cell Signaling, Danvers, MA, USA), p70 S6 Kinase (#9202, Cell Signaling), 4E-BP1 (R-113, sc-6936, Santa Cruz, Dallas, TX, USA), PARP (#9542, cell signaling), Cleaved PARP (#9541, Cell signaling), and p-H2AX (#2577, Cell Signaling). The signal was detected using HRP-conjugated secondary antibodies and Amersham ECL prime detection reagent (GE Healthcare, Chicago, IL, USA). Exposures were captured using a CCD camera imaging system (UVP; Upland, CA, USA).

### 2.6. Mitochondrial ROS Staining

MEER cells were plated on 35 mm dishes. The next day, cells were treated with the indicated treatments for one day and cells were then stained with Mito SOX (Invitrogen, Carlsbad, CA, USA) for 10 min, according to the manufacturer’s protocol, and analyzed using flow cytometry. For ROS scavenging, N-acetyl cysteine (NAC; Sigma) was used at 1 mM. The FL2 channel of an Accuri C6 flow cytometer was used for the analysis of an equal number of events for each treatment condition.

### 2.7. Lactate Assay

MEER cells were plated on 100 mm dishes and grown to 100% confluency. Media was then changed to DMEM without bicarbonate plus 10% FBS with or without the indicated treatments, and media samples were collected at 5 min and again at 4 h. Lactate assays were performed per the manufacturer’s protocol using a commercially available colorimetric lactate assay kit purchased from Eton Bioscience Inc. (San Diego, CA, USA). 

### 2.8. In Vivo Tumor Growth Assay

In vivo growth was assayed using previously described techniques [24]. All experiments were performed in accordance with institutional and national guidelines and regulations, and the protocol was approved by the animal care and use committee at Sanford Research. Briefly, using a 25-gauge needle, C57BL/6 mice (immune competent) were injected with 0.5 × 10^6^ or 1 × 10^6^ MEER cells in a volume of 100 microliters in the subcutaneous tissue of the right flank (6–12 mice per treatment condition). After 10–14 days from tumor cell injection /implantation, the mice were treated as indicated. Cisplatin was dissolved in isotonic saline and administered intraperitoneally (IP) once a week for 3 weeks. The tumor site was also irradiated at a dose of 8 Gy, with lead shielding to the rest of the body, also once weekly for three weeks (24 Gy total), on the same days as cisplatin administration. DCA and quercetin were dissolved in isotonic saline with 3% Tween 80 and were administered IP five days per week for up to 8 weeks in surviving animals at 120 mg/kg and 25 mg/kg, respectively. Animals were euthanized when the tumor size reached greater than 15 mm in its greatest dimension or when the animal became substantially emaciated. Mice were considered to be tumor-free when they showed no evidence of tumor after 3 months. Survival graphs were generated by standardizing the date of euthanization to a 2-cm tumor. Statistical analysis for the survival graphs was performed using the log-rank test, with *p* ≤ 0.01.

### 2.9. In Vitro and In Vivo pH Measurements

In vitro, MEER cells were plated on 100 mm dishes and grown to 90% confluency. Media was changed to DMEM without bicarbonate plus 10% FBS with or without the indicated treatments. Measurement of media pH was performed using a pH meter at 5 min and again at 4 h.

In vivo, tumor pH measurements were performed using C57Bl/6 mice. MEER cells were injected into the right flank of C57Bl/6 mice at 1 × 10^6^ cells per mouse. After 14 days the mice were treated with the indicated reagents IP daily for 7 days. On the 7th day of treatment, reagents were injected IP two hours before the mice were anesthetized for tumor pH measurement. The pH of the tumor was measured using a needle pH meter. The skin was retracted over the tumor site, each tumor was measured for pH at two distinct locations, and at each location, three readings were obtained at three different depths.

### 2.10. Immunohistochemistry

The paraffin-embedded tissues were sectioned at 5 μm. The BenchMark XT automated slide staining system (Ventana Medical Systems, Inc., Tucson, Arizona) was used for the optimization and staining of all antibodies. The Ventana iView DAB detection kit (Roche Diagnostics, Indianapolis, IN, USA) was used as the chromogen and the slides were counterstained with hematoxylin. Omission of the primary antibody served as the negative control. Anti-cleaved caspase-3 antibody (BioCare Medical, Concord, CA, USA) was diluted 1:100 and incubated for 1 h and 32 min, and anti-Ki67 (BioCare Medical) was diluted 1:100 and incubated for 2 h. Foxp3+ antibody (Ab20034, Abcam, Waltham, Boston, MA, USA) was diluted 1:100. The secondary antibody used is a Biotin-SP-conjugated AffiniPure Goat Anti-Rabbit IgG (H+L) diluted 1:1000 from Jackson ImmunoResearch (West Grove, PA, USA).

### 2.11. Statistical Analysis

Error bars showed SDs from the mean of at least three replicates. Two-tailed pairwise Student’s *t*-tests were conducted to compare two groups. Kaplan–Meier survival analysis and log-rank significance tests were performed with *p* values generated using the pairwise multiple comparison Holm–Sidak method. *p* values less than or equal to 0.05 were considered to have significance.

## 3. Results

### 3.1. DCA and Quercetin Inhibit mTOR and Have Synergistic Inhibitory Effects on Cell Proliferation

Our past work suggests that therapeutically modulating lactate production concurrent with standard-of-care cisplatin/radiation therapy (CRT) can increase tumor clearance in an in vivo mouse model of HPV+ HNSCC [17]. In this study, we evaluate a novel drug combination with the potential to not only decrease lactate production but also prevent cell proliferation and induce cell death. Quercetin, as described above, is readily available as a dietary supplement and has been associated with cancer prevention [38,39,40]. We combined quercetin with the better-known metabolic modulator, DCA, and assessed their effects on MEER cells, which is a previously established mouse model of HPV+ HNSCC [17]. Colony-forming assays were first performed to examine the effects of these two compounds, as well as their combination, on MEER cell clonogenic potential. MEERs were plated on 100 mm dishes and treated with DCA and quercetin for one week. DCA (5 mM) and quercetin (25 µM), individually, had significant effects on colony-forming potential, as shown in Figure 1A,B. The combination of DCA and quercetin also significantly inhibited colony formation, and to a greater extent than either compound alone.

To then determine whether the drug combination has synergistic effects, we used CalcuSyn software (version 2.1), to analyze the effects of DCA and quercetin after a series of dilutions and combinations of the two compounds. MEER cells were plated on 96-well plates and treated with either DCA or quercetin alone or in combination. After two days of treatment cells were stained using the dead cell stain, sytox green, as described in the Materials and Methods section above. Percentages of dead cells were input into CalcuSyn, and the calculated CI values are shown in Figure 1C. A CI value less than one is considered to indicate synergistic effects for two drugs. Indeed, DCA and quercetin at a ratio of 250:1 at ED 50, ED 75, and ED 90 have strong synergistic cytotoxic effects.

This synergistic growth inhibition could be related to any of several mechanisms. Quercetin has been shown to inhibit the mammalian target of rapamycin (mTOR) signaling pathway [41], a key signaling node in growth, proliferation, and metabolism. To investigate this possible explanation for the inhibition of cell proliferation, Western blotting was performed to detect the phosphorylated levels of the downstream targets of mTOR ribosomal S6 kinase (S6K) and eukaryotic initiation factor 4E binding protein 1 (4EBP1), as shown in Figure 1D. Quercetin was seen to inhibit the phosphorylation of S6K (p-S6K), and in combination with DCA, the total level of S6K was reduced. Furthermore, 4EBP1 was hypophosphorylated compared to control. Together, these data indicate that mTOR is inhibited by both quercetin alone and in combination with DCA, which could explain the growth inhibition and decreased colony size seen in quercetin and DCA-treated cells. It is the combination of DCA and quercetin, however, that synergistically has the most potent effect on inhibition of colony formation and induction of cell death, as further examined below.

### 3.2. DCA and Quercetin Induce Apoptosis

Since it was observed that DCA and quercetin synergize to inhibit cell proliferation and induce cell death, we next sought out an understanding of the mechanisms of cell death caused by the treatment combination. In Figure 2A, light microscopy imaging shows there are fewer healthy, attached cells and more floating cells with the combination treatment of DCA (5 mM) and quercetin (25 µM) after one day of treatment, suggesting that these drugs induce cell death as well as inhibit cell proliferation. Trypan blue cell exclusion assays were used to detect percentages of cell death and the results are shown in Figure 2B. Increased cell death was observed in the DCA and quercetin-treated group. Western blotting of poly-ADP ribose polymerase (PARP) indicated that there are increased levels of cleaved PARP (Figure 2C) in the combination treatment as well as increased caspase 3 cleavage (Figure 2D). This indicates that the combination treatment of DCA and quercetin induces apoptosis in MEER cells. Quercetin itself is known to cause apoptotic cell death in certain cancer cells and these results confirm this in MEER cells. The combination of DCA and quercetin shows increased levels of cleaved PARP and apoptotic cell death than either compound alone.

### 3.3. DCA and Quercetin Increase DNA Damage through Enhanced ROS Production

ROS induction is known to cause damage to DNA and lead to cell death. Since both DCA and quercetin have been shown to enhance ROS production, we hypothesized that the observed cell death caused by the combination of DCA and quercetin is associated with enhanced ROS production. We examined the effects of DCA, quercetin, and their combination on ROS production using Mito SOX staining. Both DCA and quercetin increase mitochondrial ROS production, and together they have enhanced effects on the induction of ROS, as seen in Figure 3A,B. Increased ROS production was correlated with increased phosphorylated H2AX, which is an indicator of ROS-induced DNA damage, as shown in Figure 3C. As a further demonstration, MEER cells were co-treated with N-acetyl cysteine (NAC, 10 mM), which is a ROS scavenger, and the results in Figure 3D–F show that NAC can partially prevent the DCA/quercetin increased ROS production and phosphorylation of H2AX, as well as the induction of cleaved PARP. These data indicate that the enhanced production of ROS by the combination of DCA and quercetin plays a significant role in the induction of cell death through DNA damage. This could further explain the synergistic effects of DCA and quercetin on the induction of apoptotic cell death, as shown above. However, since NAC can only partially prevent the cell death caused by the combination of DCA and quercetin, the underlying mechanism of cell death is more plausibly explained as multifactorial, and at least in part by the combination of ROS production and mTOR inhibition, as described previously (Figure 1).

### 3.4. DCA and Quercetin Prevent Extracellular Acidification

The importance of an immune response has been clearly demonstrated as necessary for clearance of HPV+ HNSCC [8], and modulation of tumor cell metabolism has been demonstrated to be a potentially safe and effective means of enhancing this immune response [17]. Our recent work and unpublished data suggest that DCA can enhance tumor clearance through reduction of lactate production and disinhibition of the immune response in vivo. As DCA and quercetin are both potent regulators of lactate synthesis and transport, we next examined the effects of DCA, quercetin, and their combination on tumor cell lactate production and extracellular acidification. MEER cells were cultured in 100 mm dishes and treated with vehicle, DCA (5 mM), quercetin (25 µM), or the combination of DCA and quercetin for 4 h. Tissue culture media samples were collected in 5 min and again in 4 h. The change in measured lactate concentration and pH are plotted in Figure 4A,B. Both DCA and quercetin as single agents decreased lactate levels compared to control, and the combination of DCA and quercetin had the most effect on attenuation of lactate production. Culture media pH was also measured, and these measurements corresponded with the changes in lactate levels in the media with the different treatments. These data confirm the role of DCA and quercetin in the regulation of metabolism via lactate production and secretion. As shown in Figure 4, both DCA and quercetin can prevent lactate production and increase the pH of the culture medium, and together they have increased effects. When also considering the synergistic effects seen on cell death induction, substantiation of this treatment combination in our in vivo mouse model was warranted to evaluate the effects of DCA, quercetin, and their combination on tumor growth and clearance.

### 3.5. DCA and Quercetin Inhibit Tumor Growth and Enhance Clearance in an Immune-Competent HPV+ HNSCC Mouse Model and This Is Associated with Decreased Percentage of Ki67 Positive Cells and Treg (+) Lymphocytes

As described in the materials and methods section, tumors were established in 40 wild-type male C57Bl/6 mice which were randomly assigned into four groups: (1) CRT, (2) CRT + DCA, (3) CRT + quercetin, and (4) CRT + DCA/quercetin. Tumor volume was measured weekly. As shown in Figure 5A, the combination of DCA and quercetin with CRT had the greatest effect on tumor growth, while DCA or quercetin individually concurrent with CRT also slightly reduced tumor growth compared with CRT alone. These differences in tumor growth correlated with survival. Survival was determined at 3 months, the predefined endpoint where animals showing no signs of tumor were declared to be tumor-free. DCA and quercetin individually concurrent with CRT increased tumor clearance compared to CRT alone, while the combination of DCA and quercetin showed the largest effect on enhancement of tumor clearance among the treatment groups. In addition to just prolonging survival, DCA, quercetin, and more so their combination, increased the number of animals who went on to be tumor-free.

Next, fixed tissue tumor specimens were examined for the proliferation marker, Ki67, to investigate possible mechanisms of the drug combination on tumor growth in vivo. As shown in Figure 5A, tumor growth was inhibited and the survival rate (Figure 5C) was increased with both DCA and quercetin treatment as well as their combination. This was associated with decreased Ki67-positive cells and increased cleaved caspase 3 staining (Figure 5D) in tumor samples.

Enhancement in growth inhibition and tumor cell death leading to increased tumor clearance by the combination of DCA and quercetin is multifactorial, as suggested by the presented data collectively. Beyond the direct cell effects of DCA and quercetin, their effects on the tumor microenvironment may also play significant roles. As both DCA and quercetin have been shown to have metabolism-modulating properties in terms of lactate production and transport, we next examined the effects of DCA and quercetin on the pH of the tumor microenvironment. Four mice per treatment group were treated with vehicle, DCA, quercetin, or DCA/quercetin. Treatments were administered for one week after tumor establishment. Tumor pH was directly measured using a pH needle electrode. As expected, DCA and quercetin as single agents were seen to lower the acidification of the tumor, and the combination of DCA and quercetin had the greatest effect, trending toward a higher pH (Figure 5E). The number of Foxp3(+) Treg cells per field in the tumor samples (Figure 5F) was reduced with treatments as well, indicating that the reduction of acidification of the tumor microenvironment is involved in the regulation of Treg infiltration.

## 4. Discussion

Our study has shown that DCA and quercetin, as single agents, have inhibitory effects on the growth of head and neck squamous cancer cells, both in vitro and in vivo. Combined, DCA and quercetin have multifaceted, synergistic antitumor effects in vitro and inhibit growth while enhancing overall clearance in an in vivo tumor mouse model of HPV+ HNSCC. The observed in vitro effects are driven by the constructive interaction of these two compounds, leading to inhibition of cell proliferation and induction of apoptotic cell death through enhanced ROS production. As for the observed in vivo effects of this novel drug combination, tumor growth inhibition, induction of cell death, and maintenance of an unfavorable tumor microenvironment, namely prevention of acidification via lactate transport, all likely play a role.

Cancer cells’ dependence on substrate-level phosphorylation as a primary metabolic pathway, both for energy and biosynthetic intermediates, provides a unique opportunity for therapeutic exploitation. This process is necessary for the generation of ATP, which necessitates regeneration of NAD+. Regeneration of NAD+ is accomplished through the fermentation of pyruvate into lactate, which is then cotransported out of the cell along with a proton down their concentration gradients, keeping the process viable. Disruption of this process not only deprives a cell of energy, but also prevents glycolysis from proceeding to intermediates necessary for the anabolic reactions of a rapidly replicating and dividing cancer cell. The result is slowed proliferation or a complete inability of the cancer cell to divide, as was seen in this study using the metabolic modulators DCA and quercetin. As cancer cells commonly adapt and compensate for therapeutic insults, and as emerging results in cancer research continue to support, inhibiting multiple targets in the same or cross-talking pathways is likely to enhance therapeutic efficacy. Our results support this notion, as DCA and quercetin combined to have the greatest growth inhibitory effects. Of course, the most significant drawback to inhibiting multiple targets in a pathway is the potential enhancement of toxicities. However, as normal cells infrequently depend heavily on glycolysis, combining multiple metabolic modulators, such as DCA and quercetin, may be a nontoxic way to enhance therapeutic efficacy.

Beyond growth-inhibitory effects, inhibition of metabolism can induce cellular outcomes through mechanisms unrelated to the depletion of energy sources and building blocks. DCA is an inhibitor of pyruvate dehydrogenase kinase (PDK). Inhibition of PDK disinhibits pyruvate dehydrogenase, promoting pyruvate entry into the TCA cycle instead of through its fermentation pathway. This not only inhibits lactate production and extracellular acidification, but also promotes oxidative phosphorylation. Passage of electrons into the electron transport chain naturally leads to increased ROS production.

Though quercetin is well known to be involved in the regulation of glucose uptake and lactate transport, its role in ROS production is less known. Our data show that quercetin, like DCA, can increase ROS production, suggesting that DCA and quercetin may synergize in this regard. When combined in this study, DCA and quercetin further enhanced ROS production and subsequent oxidative DNA damage. It is known that chemoradiation also increases ROS production, suggesting additive contributions by each component of the tested in vivo treatment combination to sufficient oxidative damage to initiate the observed programmed cell death. As DCA and quercetin were also seen to combine to inhibit cell proliferation more effectively through an apparent metabolic blockade, as indicated by attenuated lactate production and extracellular acidity and mTOR inhibition, it is likely that the enhanced tumor clearance observed in vivo is a unique mechanism whereby multiple antitumor effects of these metabolic inhibitors together enhance therapeutic efficacy.

Though not directly studied in this work, the assays of extracellular pH and lactate were geared at bringing to light, yet another potential antitumor effect imposed by DCA and quercetin and enhanced by their combination. Lactate and low pH in the tumor microenvironment contribute to many tumor-favoring processes, as explained in the introduction, and the inhibitory effects of tumor lactate production on the immune response are well documented [17]. HPV+ HNSCCs are antigenic tumors by nature of their viral protein expression. The immune response to these tumors is thus therapeutically significant, and, in fact, has been demonstrated to be induced by chemoradiation and necessary for tumor clearance [8]. Thus, intuitively, attenuating tumor lactate production has been suggested to be a means of enhancing immune cell function against HPV+ HNSCC [17]. In this previous manuscript, we demonstrated that mTOR inhibitors improve the survival of mice bearing MEER tumors through enhanced immune-mediated clearance, in part due to decreased lactate production. Both DCA and quercetin were seen to attenuate extracellular lactate levels and increase pH in vitro. 

The combination was also in increasing tumor microenvironment pH in tumors in vivo. The effects of DCA, quercetin, and their combination on tumor cell lactate production may, thus, also contribute to the antitumor efficacy observed in vivo. Taken all together, these results suggest that DCA and quercetin, both individually and better so combined, have the potential to inhibit tumor growth, enhance cell death, and help maintain an unfavorable tumor microenvironment, leading to improved survival and enhanced HPV+ tumor clearance. Further work is, however, necessary to study the lactate attenuating abilities of these drugs on immune response and the other known tumor-promoting effects of a lactate-rich, acidic tumor microenvironment.

An exciting aspect of the studied treatment combination is that DCA is already an FDA-approved drug and is currently in clinical trials for head and neck cancers [23]. Additionally, quercetin is a well-known over-the-counter dietary supplement that is readily available. Studies of DCA and quercetin could thus be rapidly translated to the clinic. Our study supports further clinical trials in patients with HPV+ head and neck cancer. Through the combination of DCA and quercetin, we have gained the knowledge that different regulators of metabolism, specifically of lactate, may work together synergistically to not only maintain an unfavorable tumor microenvironment, but also to inhibit tumor growth and enhance cell death. Taken all together, these results suggest that DCA and quercetin, both individually and better so combined, have the potential to inhibit tumor growth, enhance cell death, and help maintain an unfavorable tumor microenvironment, leading to improved survival and enhanced HPV+ tumor clearance. 

## 5. Conclusions

Here, we show that two naturally occurring compounds, DCA and quercetin, have important anti-cancer effects in head and neck cancer. Both compounds induce apoptosis and synergistically inhibit proliferation while blocking the mTOR signaling pathway in MEER cells. These two compounds increase DNA damage through enhanced ROS production, reduce tumor pH and lactate, and inhibit tumor growth. The inhibition of tumor growth is associated with enhanced clearance in this immune-competent HPV+ HNSCC mouse model, which is associated with immunosuppression involving the production of Treg (+) lymphocytes. Our study, in which the addition of quercetin with DCA suggests an enhanced synergistic effect, supports further clinical trials in patients with HPV+ head and neck tumors. Although concern regarding toxicities grows with each new drug addition to a treatment regimen, a combination like DCA and quercetin that together impose multifaceted antitumor effects may prove to be an effective therapeutic approach.

## Figures and Tables

**Figure 1 cancers-16-01525-f001:**
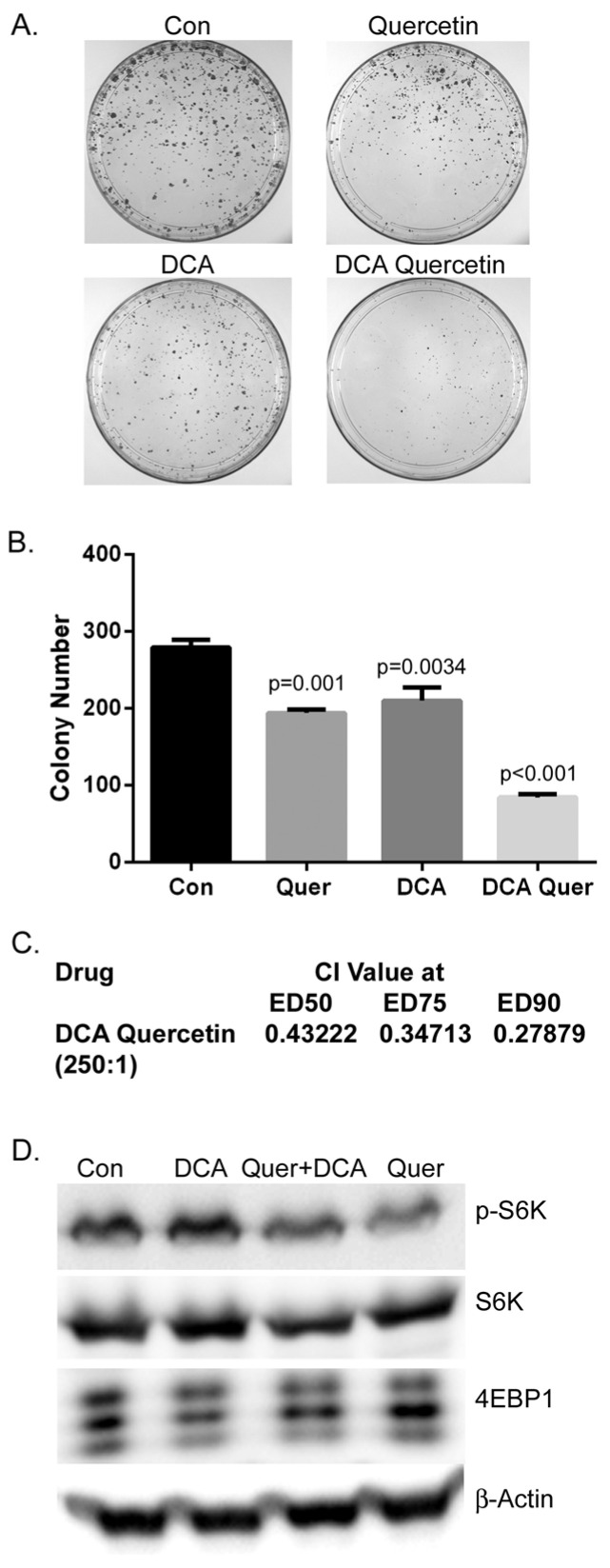
DCA and quercetin inhibit mTOR signaling and have synergistic effects on inhibition of cell proliferation. For colony formation assays, MEER cells were plated on 100 mm dishes. The next day, cells were treated with the indicated treatments for one week, followed by staining with crystal violet. Representative images are shown in (**A**), and the corresponding average colony number as well as the *p* value compared with the control is plotted in (**B**). For cell cytotoxicity assays, cells were plated on 96-well plates and treated with either DCA, quercetin, or their combination. After two days, cells were stained with sytox green nucleic acid stain, and percentages of dead cells and possible synergy were analyzed using CalcuSyn software (Version 2.1) (**C**). (**C**) CI values at ED 75 and ED 90 demonstrated synergistic effects of the combination of DCA and quercetin. Western blotting was performed for levels of downstream targets of mTOR: p-S6, total S6, and 4EBP1, with actin serving as an internal loading control, and blots are shown in (**D**). Entire original blots are shown in Appendix A.

**Figure 2 cancers-16-01525-f002:**
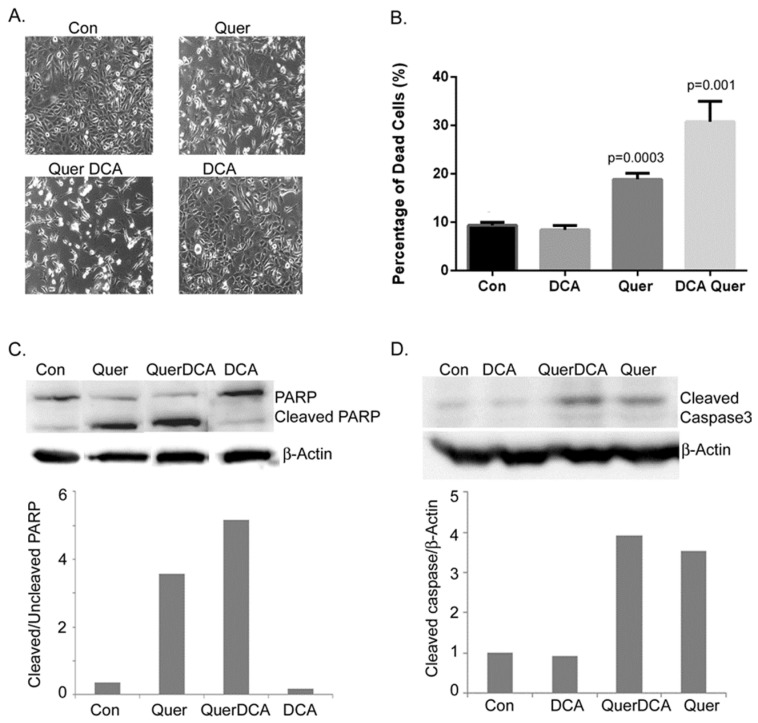
MEER cells were treated with DCA (5 mM), quercetin (25 µM), or their combination for one day. (**A**) Light microscopy images were taken at 100×. (**B**) Percentages of dead cells were determined using trypan blue cell exclusion. (**C**,**D**) Western blotting was performed for detection of PARP, cleaved PARP, and cleaved caspase 3. Actin was blotted as a loading control. Ratios of cleaved to uncleaved protein or cleaved protein to actin are plotted below their respective blots. The complete uncropped blots are shown in Appendix A.

**Figure 3 cancers-16-01525-f003:**
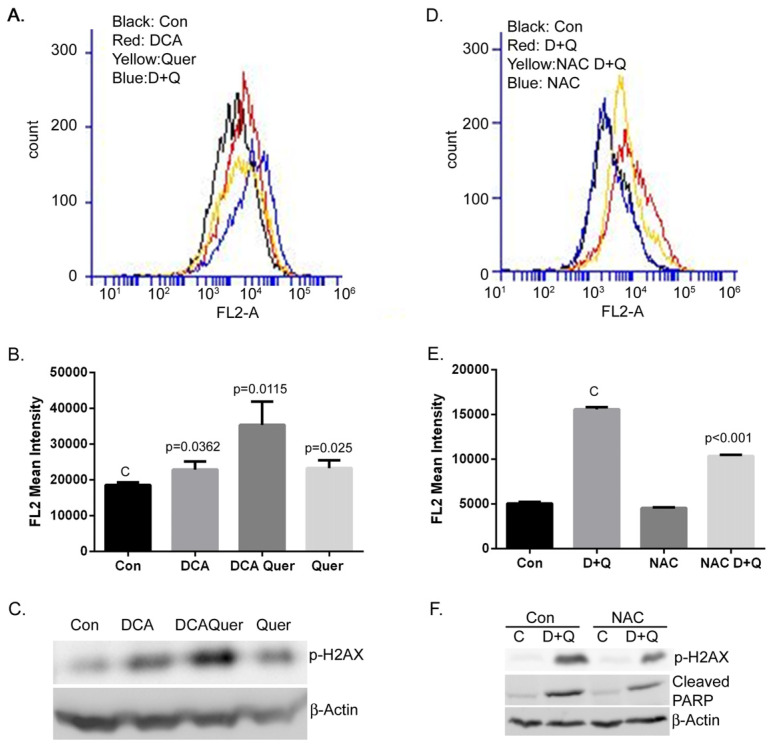
MEER cells were treated with DCA (5 mM), quercetin (25 μM), or their combination for one day. Cells were then stained with Mito SOX (10 μM) for ten minutes, trypsinized, and resuspended for flow cytometry. Representative histograms of Mito SOX staining are shown in (**A**) and mean fluorescent intensity of the replicates is shown in (**B**). Western blotting for p-H2AX showed correlative increases in DNA damage with DCA, quercetin, and more so, their combination, as shown in (**C**). When N-acetyl cysteine (10 mM), a ROS scavenger, was added to DCA/quercetin-cotreated cells, the increases in ROS observed by the drug combination were partially rescued, as shown in (**D**,**E**). Western blotting showed correlative decreases in both DNA damage and induction of apoptosis via p-H2AX and cleaved PARP, respectively, in NAC-treated cells compared with the combination treatment of DCA/quercetin alone, as shown in (**F**). The entire uncropped blots are shown in Appendix A.

**Figure 4 cancers-16-01525-f004:**
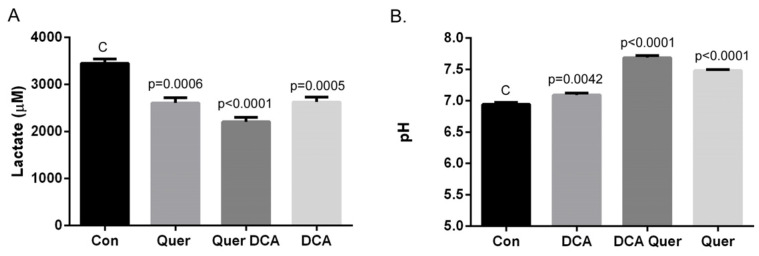
Culture media pH was measured after 4 h of the indicated treatments. (**A**) Change in pH from time zero was measured using a pH meter, and (**B**) lactate concentration in the media was measured using a commercially available lactate assay kit as described in the Materials and Methods. DCA, quercetin, and more so, their combination, increased media pH and decreased lactate concentration at 4 h. Calculated *p* values are listed above error bars in graphs, with c highlighting the control treatment.

**Figure 5 cancers-16-01525-f005:**
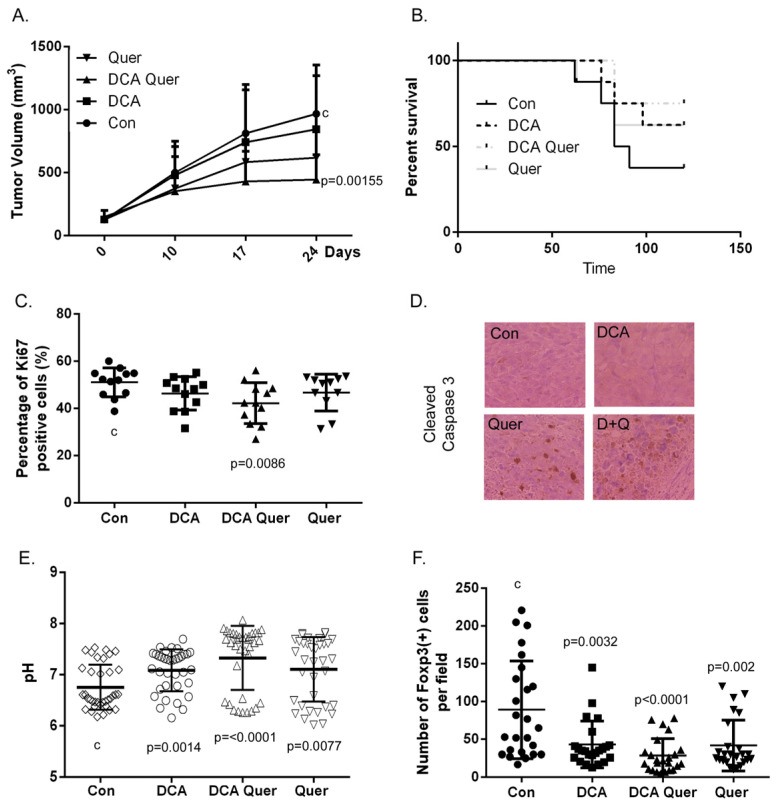
MEER cells were injected into the subcutaneous right flank of C57Bl/6 mice and treated with indicated treatments for three weeks concurrent with once weekly chemoradiation also for three weeks. Tumor volumes were monitored weekly by measuring the tumor dimensions. Growth curves are shown in (**A**), indicating that the combination of DCA/quercetin has the greatest effect on tumor growth. (**B**) Survival plot for the various groups of mice over time. (**C**) Proliferation of the cells in tumor section was illustrated by graphing the percentage of Ki67-positive cells in sections of tumor samples. (**D**) Images of cleaved caspase 3 staining on sections of tumor samples. (**E**) Tumor pH measurements were performed using mice with tumors treated for one week with the indicated treatment plus one of the three rounds of chemoradiation. (**F**) The staining of Foxp3(+) cells on tumor samples was plotted as a percentage of Foxp3(+) cells per field.

## Data Availability

The original data from this study are included in the text of article or Appendix A. Further inquiries can be addressed to corresponding authors.

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
