# Peer review of "Dichloroacetate and Quercetin Prevent Cell Proliferation, Induce Cell Death and Slow Tumor Growth in a Mouse Model of HPV-Positive Head and Neck Cancer"

_cancers, 2024, doi:10.3390/cancers16081525_

Round 1

Reviewer 1 Report

Comments and Suggestions for Authors

The research study by Zhuang et al, “DCA and quercetin prevent cell proliferation, induce cell death, and slow tumor growth in a mouse model of HPV-positive head and neck cancer” investigated the anticancer efficacy of naturally occurring plant flavonoids [Dichloroacetate (DCA) and quercetin] combination in both in vitro and in vivo studies. The authors demonstrated their inhibitory effect at the cellular level, manifesting in reduced cell viability, diminished colony formation, and increased apoptosis. Additionally, the research indicated that the combined administration of DCA and quercetin led to a reduction in tumor growth and improved survival in immune-competent mice. However, the study's overall significance is compromised by a lack of comprehensive information regarding the synergistic effects of DCA and quercetin in MEER cells and mice studies, raising noteworthy concerns that warrant further attention and clarification.

Major Comments:

1.     The authors evaluated the anticancer potential of DCA and quercetin in conjunction with cisplatin/radiation therapy in a murine model. While the study extensively examined the impact of DCA and quercetin at the cellular level, it did not compare the efficacy between DCA and quercetin alone versus their combination with cisplatin/radiation therapy. To establish the comprehensive anticancer efficacy of DCA and quercetin, it is recommended that the authors either add references of prior studies that have explored this specific aspect or provide relevant evidence about the DCA and quercetin combination group to fortify the significance of their findings. This addition would enhance the overall impact and visibility of the study.

2.     The authors presented the outcomes of cell death percentage, cleaved PARP, and cleaved caspase 3 in Figure 2 following treatment with DCA and quercetin alone, as well as in combination. The results indicated that DCA, when given independently, was ineffective; however, in combination with quercetin, a significant increase was observed in the percentage of dead cells, cleaved PARP, and cleaved caspase 3. To enhance the clarity of the study, it is suggested to expand on the potential reasons for this synergistic effect when DCA and quercetin are administered concurrently. The inclusion of such insights will provide a more comprehensive understanding of the observed outcomes. Additionally, the reviewer is interested in whether the authors explored various time points to assess the inhibitory effect of this combination on MEER cells. Including information on different time points would contribute valuable insights into the treatment response and strengthen the robustness and thoroughness of the study.

3.     The author should incorporate the material and method section and address the findings in the result and discussion sections of the manuscript concerning the data presented in Figure 6.

Minor Comments:

1.     The author should clarify the criteria used to select DCA (5 mM) and quercetin (25 µM) for the cellular study.

2.     Please improve the image quality of all the figures, particularly microscopic images that are difficult to understand.

3.     Authors should also include data on weight changes post-DCA and quercetin administration, considering their metabolic modulating properties. Additionally, a histopathological evaluation of tumor tissue could provide valuable insights into the adjuvant therapy effect of DCA and quercetin.

4.     Please include a scheme for animal study with treatment and their timelines.

5.     Please include the full abbreviation of DCA in the title.

6.     Please specify the name of the cells and the volume administered to mice during the in vivo tumor growth assay.

7.     Please modify the symbol for the treatment groups in Figure 5A to improve differentiation between the groups, as the current symbols are challenging to discern.

8.     In Figure 5D, the author should incorporate a graph representing the data on cleaved caspase 3 in tumors. Additionally, include representative images for Ki67 in Figure 5C.

Author Response

                                                                                                                                                                                         January 11, 2024

Dear Editor,

We have responded to the comments made by reviewers. We have included the information you requested prior to discussing the responses to the Reviewers.

Major Comments:

  1. The authors evaluated the anticancer potential of DCA and quercetin in conjunction with cisplatin/radiation therapy in a murine model. While the study extensively examined the impact of DCA and quercetin at the cellular level, it did not compare the efficacy between DCA and quercetin alone versus their combination with cisplatin/radiation therapy. To establish the comprehensive anticancer efficacy of DCA and quercetin, it is recommended that the authors either add references of prior studies that have explored this specific aspect or provide relevant evidence about the DCA and quercetin combination group to fortify the significance of their findings. This addition would enhance the overall impact and visibility of the study.

Reply: Thank you for the suggestions, the response to cisplatin/radiation in this murine model has been shown to mimic clinical response in patients with HPV+ head and neck cancer. Since the goal of this study was to determine whether any treatment can further improve the response of cisplatin/radiation, and not as stand-alone treatment option, the anticancer potential of DCA and quercetin was evaluated in combination with standard of care: cisplatin/radiation, not tested alone. The extensive work in vitro is to demonstrate that the combination of DCA and quercetin does have effects on further increasing cell death in vitro and has the potential to improve the tumor environment as they modify the tumor metabolism. It is not advised to consider DCA and quercetin treatment without combination with standard-of Care treatment despite the promising in vitro observations.

  1. The authors presented the outcomes of cell death percentage, cleaved PARP, and cleaved caspase 3 in Figure 2 following treatment with DCA and quercetin alone, as well as in combination. The results indicated that DCA, when given independently, was ineffective; however, in combination with quercetin, a significant increase was observed in the percentage of dead cells, cleaved PARP, and cleaved caspase. To enhance the clarity of the study, it is suggested to expand on the potential reasons for this synergistic effect when DCA and quercetin are administered concurrently. The inclusion of such insights will provide a more comprehensive understanding of the observed outcomes. Additionally, the reviewer is interested in whether the authors explored various time points to assess the inhibitory effect of this combination on MEER cells. Including information on different time points would contribute valuable insights into the treatment response and strengthen the robustness and thoroughness of the study.

Reply: Thanks for suggestions. Both quercetin and DCA have been shown to increase ROS levels in vitro. Our data suggested that DCA alone did not have a significant r ole in modulating cell death in MEER cells despite increased ROS production. DCA has been shown to increase reactive oxygen species (ROS) production, lead to cancer cell death (20, 21) in other studies. It is likely that MEER cells have compensatory mechanisms to overcome the increased ROS production when the cells were only treated with DCA. Quercetin itself was able to increase cell death and ROS production, therefore, the further increase of ROS by combining with DCA could contribute to the increased ROS and cell death in the combination treatment. In our previous study (reference [17] in manuscript),we showed that rapamycin inhibits mTOR metabolism, decreases lactate, and enhances the immune mediated clearance of tumor cells. Both DCA (through inhibition of pyruvate dehydrogenase kinase) and quercetin (through inhibition of glucose uptake and lactate secretion) also lower tumor lactate. Thus, we hypothesized that combining quercetin with DCA would result in more efficacious reduction of lactate excretion in the tumor microenvironment, leading to enhanced immune-mediated clearance following chemoradiation. Phase II clinicals trials demonstrated that DCA in combination with chemoradiotherapy for head and neck squamous cell carcinoma [Powell, 2022] was safe with no detrimental effects on survival.

    3. The author should incorporate the material and method section and   address the findings in the result and discussion sections of the manuscript concerning the data presented in Figure 6.

Response: The materials and methods were edited to explain the data in Fig.6 and the first paragraph of the discussion involves the data in Fig.6.

Minor Comments:

  1. The author should clarify the criteria used to select DCA (5 mM) and quercetin (25 µM) for the cellular study.

Concentrations of DCA was determined by analysis of the manuscript Lucido et al., 2018. and concentration of quercetin was determined based on prior experiments in the lab. Furthermore, it is known that DCA reaches mM levels in vivo.

  1. Please improve the image quality of all the figures, particularly microscopic images that are difficult to understand.

The images submitted as TIF are clearer than the figures which were imported into Microsoft word, as reviewer had Microsoft word document with inserted figures. For the final submission, only high resolution TIF files will be submitted as figures.

  1. Authors should also include data on weight changes post-DCA and quercetin administration, considering their metabolic modulating properties. Additionally, a histopathological evaluation of tumor tissue could provide valuable insights into the adjuvant therapy effect of DCA and quercetin.

No significant weight changes occurred following DCA and quercetin administration. Further studies involve histopathological evaluation of tumor tissue following adjuvant therapy with DCA and quercetin.

  1. Please include a scheme for animal study with treatment and their timelines.

The description of the animal study is described in detail in the Materials and Methods section describing in vivo tumor experiments on page 11 of the manuscript.

Implantation---10-14 days----Cisplatin/Radiation (qwx3)                                                                                                                                            Cisplatin/Radiation (qwx3) + DCA (qdx5/week for 8 weeks)                                                                                                                                        Cisplatin/Radiation (qwx3) + Quercetin (qdx5/week for 8 weeks)                                                                                                                             Cisplatin/Radiation (qwx3) + Quercetin/DCA (qdx5/week for 8 weeks)

  1. Please include the full abbreviation of DCA in the title.                                                                                                                   Reply: Title has been revised to include full name of DCA.
  2. Please specify the name of the cells and the volume administered to mice during the in vivo tumor growth assay Reply: This information has been added to the Materials and Methods.

Reviewer #2:

In the manuscript entitled "DCA and quercetin prevent cell proliferation, induce cell death, and slow tumor growth in a mouse model of HPV-positive head and neck cancer", the authors demonstrated that the combination of dichloroacetate (DCA) and quercetin synergistically reduced cell proliferation via inhibiting mTOR and increased apoptosis through enhanced ROS production in HPV-positive head and neck cancer cells. Moreover, the combination of dichloroacetate (DCA) and quercetin can meanwhile decreased acidification of the tumor microenvironment. The work presented is potentially interesting and convincing. several minor revisions are list below.

  1. Figure 1A, 2A and 5D are very unclear, please replace them in the clearer figures, better color images.

The original TIF images we will submit in final version are clearer than the figures which the reviewer saw in the Microsoft word document with contained the figures.

     2.The order of the treatments in figures is inconsistent. Please rearrange them to maintain a consistent sequence.

       Reply: The order of the treatments has been rearranged to be more consistent in Figure 1.                                                

  1. Figure 2, when referring to apoptosis, better to detect the cells stained with Annexin V and PI via flow.

       Reply: Annexin V and PI via flow is ideal for quantifying apoptotic cell death. However, cleaved caspase and PARP are also accepted methods for analysis of apoptosis and clearly show that apoptotic cell death is increased by the treatments and combinations.

  1. Figure 5A, the tumor volumes range from 500 to 1000. And it is very unusual the mice could survive around 3 months but did not reach ethics.

Reply: As those studies were performed in combination with chemo/radiation, some of the mice had tumor volumes from 500 to 1000 because the treatment was effective so that the mice that did not reach the euthanization criteria at three-month time point.

In addition to addressing the comments from Reviewers, we have addressed the points the editor discussed. Specifically, the source of cell lines used was added to the Materials and Methods. The following information added to the Materials and Methods:  “ Mouse MEER cells were provided by Dr. John Lee and previously internally derived from C57Bl/6 mouse oropharyngeal epithelium (MOE) through retroviral transduction as previously described (38).” Human cell lines were acquired from ATCC. The quantification of the western blot in Figure 1 is included as Supplementary Figure 1.

The institutional email addresses for all authors are included:[email protected]; [email protected]; [email protected]; [email protected], [email protected] , &[email protected]

Reviewer 2 Report

Comments and Suggestions for Authors

In the manuscript entitled "DCA and quercetin prevent cell proliferation, induce cell death, and slow tumor growth in a mouse model of HPV-posi- tive head and neck cancer", the authors demonstrated that the combination of dichloroacetate (DCA) and quercetin synergistically reduced cell proliferation via inhibiting mTOR and increased apoptosis through enhanced ROS production in HPV-positive head and neck cancer cells. Moreover, the combination of dichloroacetate (DCA) and quercetin can meanwhile decreased acidifcation of the tumor microenvironment. The work presented is potentially interesting and generally convincing. several minor revisions are list below.

1.      Figure 1A, 2A and 5D are very unclear, please replace them in the clearer figures, better color images.  

2.      The order of the treatments in figures is inconsistent. Please rearrange them to maintain a consistent sequence.

3.      Figure 2C, only one WB?

4.      Figure 2, when referring to apoptosis, better to detect the cells stained with Annexin V and PI via flow.

5.      Figure 5A, the tumor volumes range from 500 to 1000. And it is very unusual the mice could survive around 3 months but did not reaech ethics.

Author Response

Reviewer #2:

In the manuscript entitled "DCA and quercetin prevent cell proliferation, induce cell death, and slow tumor growth in a mouse model of HPV-positive head and neck cancer", the authors demonstrated that the combination of dichloroacetate (DCA) and quercetin synergistically reduced cell proliferation via inhibiting mTOR and increased apoptosis through enhanced ROS production in HPV-positive head and neck cancer cells. Moreover, the combination of dichloroacetate (DCA) and quercetin can meanwhile decreased acidification of the tumor microenvironment. The work presented is potentially interesting and convincing. several minor revisions are list below.

  1. Figure 1A, 2A and 5D are very unclear, please replace them in the clearer figures, better color images.

The original TIF images we will submit in final version are clearer than the figures which the reviewer saw in the Microsoft word document with contained the figures.

     2.The order of the treatments in figures is inconsistent. Please rearrange them to maintain a consistent sequence.

       Reply: The order of the treatments has been rearranged to be more consistent in Figure 1.                                                

  1. Figure 2, when referring to apoptosis, better to detect the cells stained with Annexin V and PI via flow.

       Reply: Annexin V and PI via flow is ideal for quantifying apoptotic cell death. However, cleaved caspase and PARP are also accepted methods for analysis of apoptosis and clearly show that apoptotic cell death is increased by the treatments and combinations.

  1. Figure 5A, the tumor volumes range from 500 to 1000. And it is very unusual the mice could survive around 3 months but did not reach ethics.

Reply: As those studies were performed in combination with chemo/radiation, some of the mice had tumor volumes from 500 to 1000 because the treatment was effective so that the mice that did not reach the euthanization criteria at three-month time point.

In addition to addressing the comments from Reviewers, we have addressed the points the editor discussed. Specifically, the source of cell lines used was added to the Materials and Methods. The following information added to the Materials and Methods:  “ Mouse MEER cells were provided by Dr. John Lee and previously internally derived from C57Bl/6 mouse oropharyngeal epithelium (MOE) through retroviral transduction as previously described (38).” Human cell lines were acquired from ATCC. 

The institutional email addresses for all authors are included:[email protected]; [email protected]; [email protected]; [email protected], [email protected] , &[email protected]

Round 2

Reviewer 1 Report

Comments and Suggestions for Authors

The authors have made changes as per the reviewer's suggestions and added the required information to improve the manuscript overall.

Minor comment:

It is suggested to relocate Figure 6 to the supplementary materials and mention details and the figure in the DCA and quercetin induce apoptosis result section, as its current placement in the discussion section seems disconnected.

Author Response

  1. The authors evaluated the anticancer potential of DCA and quercetin in conjunction with cisplatin/radiation therapy in a murine model. While the study extensively examined the impact of DCA and quercetin at the cellular level, it did not compare the efficacy between DCA and quercetin alone versus their combination with cisplatin/radiation therapy. To establish the comprehensive anticancer efficacy of DCA and quercetin, it is recommended that the authors either add references of prior studies that have explored this specific aspect or provide relevant evidence about the DCA and quercetin combination group to fortify the significance of their findings. This addition would enhance the overall impact and visibility of the study.

Reply: Thank you for the suggestions, the response to cisplatin/radiation in this murine model has been shown to mimic clinical response in patients with HPV+ head and neck cancer. Since the goal of this study was to determine whether any treatment can further improve the response of cisplatin/radiation, and not as stand-alone treatment option, the anticancer potential of DCA and quercetin was evaluated in combination with standard of care: cisplatin/radiation, not tested alone. The extensive work in vitro is to demonstrate that the combination of DCA and quercetin does have effects on further increasing cell death in vitro and has the potential to improve the tumor environment as they modify the tumor metabolism. It is not advised to consider DCA and quercetin treatment without combination with standard-of Care treatment despite the promising in vitro observations.

  1. The authors presented the outcomes of cell death percentage, cleaved PARP, and cleaved caspase 3 in Figure 2 following treatment with DCA and quercetin alone, as well as in combination. The results indicated that DCA, when given independently, was ineffective; however, in combination with quercetin, a significant increase was observed in the percentage of dead cells, cleaved PARP, and cleaved caspase. To enhance the clarity of the study, it is suggested to expand on the potential reasons for this synergistic effect when DCA and quercetin are administered concurrently. The inclusion of such insights will provide a more comprehensive understanding of the observed outcomes. Additionally, the reviewer is interested in whether the authors explored various time points to assess the inhibitory effect of this combination on MEER cells. Including information on different time points would contribute valuable insights into the treatment response and strengthen the robustness and thoroughness of the study.

Reply: Thanks for suggestions. Both quercetin and DCA have been shown to increase ROS levels in vitro. Our data suggested that DCA alone did not have a significant r ole in modulating cell death in MEER cells despite increased ROS production. DCA has been shown to increase reactive oxygen species (ROS) production, lead to cancer cell death (20, 21) in other studies. It is likely that MEER cells have compensatory mechanisms to overcome the increased ROS production when the cells were only treated with DCA. Quercetin itself was able to increase cell death and ROS production, therefore, the further increase of ROS by combining with DCA could contribute to the increased ROS and cell death in the combination treatment. In our previous study (reference [17] in manuscript),we showed that rapamycin inhibits mTOR metabolism, decreases lactate, and enhances the immune mediated clearance of tumor cells. Both DCA (through inhibition of pyruvate dehydrogenase kinase) and quercetin (through inhibition of glucose uptake and lactate secretion) also lower tumor lactate. Thus, we hypothesized that combining quercetin with DCA would result in more efficacious reduction of lactate excretion in the tumor microenvironment, leading to enhanced immune-mediated clearance following chemoradiation. Phase II clinicals trials demonstrated that DCA in combination with chemoradiotherapy for head and neck squamous cell carcinoma [Powell, 2022] was safe with no detrimental effects on survival.

The author should incorporate the material and method section and address the findings in the result and discussion sections of the manuscript concerning the data presented in Figure 6.

The materials and methods were edited to explain the data in Fig.6 and the first paragraph of the discussion involves the data in Fig.6.

Minor Comments:

  1. The author should clarify the criteria used to select DCA (5 mM) and quercetin (25 µM) for the cellular study.

Concentrations of DCA was determined by analysis of the manuscript Lucido et al., 2018. and concentration of quercetin was determined based on prior experiments in the lab. Furthermore, it is known that DCA reaches mM levels in vivo.

  1. Please improve the image quality of all the figures, particularly microscopic images that are difficult to understand.

The images submitted as TIF are clearer than the figures which were imported into Microsoft word, as reviewer had Microsoft word document with inserted figures. For the final submission, only high resolution TIF files will be submitted as figures.

  1. Authors should also include data on weight changes post-DCA and quercetin administration, considering their metabolic modulating properties. Additionally, a histopathological evaluation of tumor tissue could provide valuable insights into the adjuvant therapy effect of DCA and quercetin.

No significant weight changes occurred following DCA and quercetin administration. Further studies involve histopathological evaluation of tumor tissue following adjuvant therapy with DCA and quercetin.

  1. Please include a scheme for animal study with treatment and their timelines.

The description of the animal study is described in detail in the Materials and Methods section describing in vivo tumor experiments on page 11 of the manuscript.

Implantation---10-14 days----Cisplatin/Radiation (qwx3)                                        Cisplatin/Radiation (qwx3) + DCA (qdx5/week for 8 weeks)                                     Cisplatin/Radiation (qwx3) + Quercetin (qdx5/week for 8 weeks)                             Cisplatin/Radiation (qwx3) + Quercetin/DCA (qdx5/week for 8 weeks)

  1. Please include the full abbreviation of DCA in the title.                                Reply: Title has been revised to include full name of DCA.
  2. Please specify the name of the cells and the volume administered to mice during the in vivo tumor growth assay Reply: This information has been added to the Materials and Methods.
